# No MCMC Teaching For me: Learning Energy-Based Models via Diffusion Synergy

## Abstract

Markov chain Monte Carlo (MCMC) sampling-based maximum likelihood estimation is a standard approach for training Energy-Based Models (EBMs). However, its effectiveness and training stability in high-dimensional settings remain thorny issues due to challenges like mode collapse and slow mixing of MCMC. To address these limitations, we introduce a novel MCMC teaching-free learning framework that jointly trains an EBM and a diffusion-based generative model, leveraging the variational formulation of divergence between time-reversed diffusion paths. In each iteration, the generator model is trained to align with both the empirical data distribution and the current EBM, bypassing the need for biased MCMC sampling. The EBM is then updated by maximizing the likelihood of the synthesized examples generated through a diffusion generative process that more accurately reflects the EBM's distribution. Moreover, we propose a novel objective function that further improves EBM learning by minimizing the discrepancy between the EBM and the generative model. Our proposed approach enhances training efficiency and overcomes key challenges associated with traditional MCMC-based methods. Experimental results on generative modeling and likelihood estimation demonstrate the superior performance of our method.

## 1 Introduction

Energy-based models (EBMs) are an appealing class of probabilistic models that can model the data distributions and generate various types of samples, such as images (LeCun et al., 2006a; Kim & Bengio, 2016), reinforcement learning (Haarnoja et al., 2017; Boney et al., 2019), language (Mireshghallah et al., 2022), and out-of-distribution detection (Liu et al., 2020). The maximum likelihood estimation (MLE) based on Markov chain Monte Carlo (MCMC) is widely used to learn EBMs. However, training EBMs on high-dimensional data is still a challenging problem since MCMC is prone to mode collapse and slow mixing in high dimensions.

Recently, a line of work has been proposed to improve the learning of EBM by accelerating the MCMC sampling process based on MCMC teaching. In MCMC teaching, a complementary generator model is cooperatively learned to serve as the amortized initialization of MCMC (Xie et al., 2022). Specifically, the synthesized samples generated by the generative model are revised by a short-run MCMC toward the current EBM. Then, the refined synthesized samples serve as fair samples to update the EBM and the generator by maximizing the likelihood of each model. Although the MCMC revision process in MCMC teaching provides a useful gradient information for updating the synthesized examples, the non-convergent nature of short-run MCMC typically generates biased samples, leading to inaccurate estimations for updating generators and EBMs . To mitigate the bias of short-run MCMC when updating the generator, Cui & Han (2023) proposed to train the generator to match both the empirical data distribution and the EBM via dual-MCMC teaching. However, the biased short-run MCMC can still lead to conventional biased training of the generator and the EBM.

In this paper, we propose a novel MCMC teaching-free **diff**usion-based **EBM** (DiffEBM) learning framework to remove the bias of the short-run MCMC by jointly training an EBM and a diffusion-based generative model. In DiffEBM, we introduce an objective function to train the EBM and the generator to match with each other without MCMC revisions by leveraging the variational formulation of divergence of the path measures between the inference process and the generative process of probabilistic diffusion models (Richter & Berner, 2024). Training the EBM and the generator to

match each other implies that we can directly regard the samples from the generator as fair samples from the current EBM and avoid biased short-run MCMC revisions when updating both models. In such a way, our method does not suffer from the bias issue of short-run MCMC revision of the current MCMC teaching framework. Specifically, the generator is trained by maximizing the likelihood of the empirical data and minimizing the difference with the current EBM. The EBM is then learned by maximizing the likelihood of the empirical data where the synthesized examples are simulated from the diffusion model within finite time and minimizing the discrepancy with the generator. Therefore, we can avoid the non-convergent MCMC during the iterative joint learning process.

In summary, our innovations are:

- We propose an MCMC teaching-free framework to learn energy-based models based on the variational formulation of divergence between forward and backward time-reversed diffusion paths.

- Our method can avoid mode collapse and slowing mixing issues of traditional MCMC sampling-based EBM learning methods.

- We demonstrate that our method achieves significant improvements in sample quality compared to existing EBM learning approaches on benchmark simulation tasks.

## 2 RELATED WORK

**Training Energy-based models**    Given the computational costs of MCMC inference until convergence, numerous methods have been proposed to improve the learning of EBMs. For example, (Xie et al., 2018; 2016; 2020) proposed amortizing MCMC sampling with learned networks by leveraging the cooperative learning scheme. In (Xie et al., 2022), the authors train an energy-based model with a normalizing flow as an amortized sampler to initialize the MCMC chains of the energy-based model. Carbone et. al. (Carbone et al., 2023) point out that the approximations of the estimate of the gradient of the cross entropy are uncontrolled and known to induce biases similar to those observed with score-based methods and cannot handle well multimodal distributions. Therefore, they proposed a sequential Monte Carlo sampling procedure with Jarzynski correction to estimate the cross-entropy and its gradient with theoretical guarantees.

**Diffusion probabilistic models for Energy-based models**    Diffusion models (Sohl-Dickstein et al., 2015; Song et al., 2020; Ho et al., 2020) learn to reverse a constructed noising process to generate high-quality data, which inspires a line of works to improve the training of EBM on generative modeling. (Gao et al., 2021) proposed to learn a sequence of EBMs for the marginal distributions of the diffusion process, where each EBM is learned with recovery likelihoods that are defined as the conditional distributions of the reverse process. Zhu et al. (2023) introduced the cooperative diffusion recovery likelihood that jointly trained EBMs and initializer models at multiple noise levels to improve sample quality and generation performance, bridging the gap between EBMs and other generative models like GANs and diffusion models. (Luo et al., 2023) proposed a novel diffusion contrastive divergence, which replaces MCMC sampling in contrastive divergence with diffusion processes, offering a more computationally efficient training method for EBMs that avoids the difficulties of handling non-negligible gradient terms.

**Diffusion probabilistic models for sampling**    The goal of sampling is to generate samples from a known (unnormalized) density function (Liu & Wang, 2016; Cheng et al., 2023). Richter et al. (Richter & Berner, 2024) propose a general framework for learning a diffusion model tailored to the sampling task. Their approach involves minimizing the divergence between the diffusion sampling process and the time-reversal of the noising process. Sampling has a close connection to EBM learning since training EBM via MLE requires sampling from the current energy model. Our work is inspired by the diffusion-based sampling method.

# 3 PRELIMINARIES

This section provides a basic introduction to energy-based models, probabilistic diffusion models and diffusion-based sampling. A comprehensive introduction to training energy-based models can be found in (Song & Kingma, 2021).

**Notation.** In this paper, we denote $\mathcal{X}$ as the $D$-dimensional Euclidean space $\mathbb{R}^D$. Let $\mathcal{P}(\mathcal{X})$ represent the sets of probability measures on $\mathcal{X}$. For suitable functions $\psi \in C(\mathbb{R}^D \times [0, T], \mathbb{R}^D)$ and $h \in C(\mathbb{R}^D \times [0, T], \mathbb{R})$, the stochastic and deterministic integrals are defined as

$$D_h(X) \coloneqq \int_0^T h(X_t, t) \, \mathrm{d}t \qquad \text{and} \qquad S_\psi(X) \coloneqq \int_0^T \psi(X_t, t) \cdot \mathrm{d}W_t, \tag{1}$$

where $W$ is a standard $D$-dimensional Winner process.

## 3.1 ENERGY-BASED MODELS

Energy-based models (EBMs) are probabilistic models used to capture complex distributions (LeCun et al., 2006b), which rely on an energy function $E_\theta(x)$. The probability distribution of a data point $x$ under an EBM is inversely proportional to the exponential of the negative energy and can be expressed using the Boltzmann distribution as:

$$p_\theta(x) = \frac{1}{Z(\theta)} \exp\{-E_\theta(x)\}, \tag{2}$$

where $Z(\theta)$ is the intractable normalization constant.

EBMs can be learned by minimizing the Kullback-Leibler (KL) divergence of the empirical data distribution $p_{\text{data}}$ and the model distribution $p_\theta$, which is defined as

$$\arg\min_\theta \mathcal{L}_{\text{MLE}}(\theta) = \arg\min_\theta D_{\text{KL}}(p_{\text{data}} \| p_\theta) = \arg\min_\theta \left\{ E_{p_{\text{data}}} [E_\theta(X)] + \log Z(\theta) \right\}.$$

The gradient for updating $\theta$ is given by:

$$\begin{aligned} \frac{d}{d\theta} \mathcal{L}_{\text{MLE}}(\theta) &= \frac{d}{d\theta} E_{q_{\text{data}}} [E_\theta(x)] - E_{p_\theta} \left[ \frac{\partial}{\partial \theta} E_\theta(X) \right] \\ &\approx \frac{\partial}{\partial \theta} \left( \frac{1}{n} \sum_{i=1}^n E_\theta(x_i^+) - \frac{1}{m} \sum_{i=1}^m E_\theta(x_i^-) \right), \end{aligned} \tag{3}$$

where $\{x_i^+\}_{i=1}^n$ represents independent and identically distributed (i.i.d.) samples from the data distribution $q_{\text{data}}$ and $\{x_i^-\}_{i=1}^m$ are i.i.d. synthesized samples from the current learned EBM ($p_\theta$).

Sampling from the current learned distribution can be accomplished by using the following Langevin dynamics-based MCMC sampling method (Welling & Teh, 2011):

$$x^{k+1} = x^k - \alpha \nabla_x E_\theta(x^k) + \sqrt{2\alpha} z^k, \quad z^k \sim \mathbb{N}(0, \mathbb{I}), \tag{4}$$

where $\alpha$ is a predefined stepsize and $x^0$ is sampled from a standard Gaussian distribution as the MCMC initializer.

## 3.2 DIFFUSION MODELS FOR GENERATIVE MODELING

Diffusion models (Ho et al., 2020; Song et al., 2020) learn the data distribution by first perturbing the data into an isotropic Gaussian distribution and then reversing this process to generate new samples. The forward inference and backward generative process can be characterized by the following two stochastic differential equations (SDE),

$$\mathrm{d}X_t = \mathbf{f}(X_t, t)\mathrm{d}t + g(t)\mathrm{d}\mathbf{w}, \quad X_0 \sim p_{\text{data}} \tag{5}$$

$$\mathrm{d}X_t = \left[ \mathbf{f}(X_t, t) - g^2(t) \nabla_X \log p_t(X) \right] \mathrm{d}t + g(t)\mathrm{d}\mathbf{w}, \quad X_T \sim p_{\text{prior}} \tag{6}$$

where $\mathbf{w}$ is the Brownian motion, $t \in [0, T]$, $\mathbf{f}(\cdot, t)$ is a drift coefficient and $g(\cdot)$ is a diffusion coefficient. An affine $\mathbf{f}(X_t, t) = -\frac{1}{2}\beta(t)X_t$ and $g(t) = \sqrt{\beta(t)}$ lead to the variance preserving (VP) SDE, where $\beta(t) : [0, T] \to (0, 1)$ is a variance schedule.

Denoising score matching objective is widely used for estimating the score $\nabla_X \log p_t(X)$ that is required by the backward generative process, i.e.,

$$\min \mathcal{L}_{\mathrm{DSM}}(\phi) = \mathbb{E}_t \, \mathbb{E}_{p_t(X_t)} \left[ \| \mathbf{s}_\phi(X_t, t) - \nabla_{X_t} \log p_t(X_t) \|_2^2 \right]. \tag{7}$$

When the score function is learned, we can adopt the backward SDE in Equation 6 for generative modeling.

## 3.3 DIFFUSIONS MODELS FOR SAMPLING

The task of the diffusion-based sampling aims to learn to sample from the known target density $p_{\mathrm{target}} = \rho/Z$ by transporting samples from a prior density $p_{\mathrm{prior}}$ via a learnable controlled stochastic process $X^{\mathbf{u}}$ that is defined as

$$\mathrm{d}X_t^{\mathbf{u}} = \left[ \mathbf{f}(X_t^{\mathbf{u}}, t) - g(t)\mathbf{u}_\phi(X_t^{\mathbf{u}}, t) \right] \mathrm{d}t + g(t)\mathrm{d}\mathbf{w}, \quad X_T^{\mathbf{u}} \sim p_{\mathrm{prior}}, \tag{8}$$

where the process $X^{\mathbf{u}}$ is governed by a learnable control function $\mathbf{u}_\phi(X_t, t)$ with parameter $\phi$ and the desired form of $\mathbf{u}_\phi(X_t, t)$ is the score function of the time-reversal of the reference process defined in Equation 5 that transfers the noise into data under the target density.

Diffusion-based sampling differs from the generative modeling setting since there is no empirical data available to calculate the noised version of data for estimating the control (score) function via score matching methods. When only the density function is available for the sampling task, the control function can be learned based on the variational formulation of divergence of the path measures between the inference process and the generative process of probabilistic diffusion models. To compute such a tractable divergence, Richter & Berner (2024) derived the following formulation of the likelihood of path measures between the generative process and the time-reversal of the inference process that is related to the unnormalized target density.

Let $P_{\mathcal{X}^{\mathbf{u}}}$ denote the path space measure of the process $X^{\mathbf{u}}$ and define $P_{\tilde{\mathcal{X}}}$ as the path space measure of the time-reversal of $\mathcal{X}$. To estimate the control function $\mathbf{u}_\phi(X_t, t)$ based on the known unnormalized density function, the likelihood of path measures between the generative process $X^{\mathbf{u}}$ and the target process $P_{\tilde{\mathcal{X}}}$ is defined as the following Radon-Nikodym derivative (Richter & Berner, 2024)

$$\frac{\mathrm{d}P_{\mathcal{X}^{\mathbf{u}}}}{\mathrm{d}P_{\tilde{\mathcal{X}}}}(\mathcal{X}^\beta) = Z \exp\left( D_{h_{u,\beta}} + S_u - R \right)(X^\beta), \tag{9}$$

where $X^\beta$ is a reference process as defined in Equation 8 by replacing the function $\mathbf{u}$ with $\beta$, and the integral functions $D$ and $R$ are defined in Equation 1, and $\quad R(X^\beta) := \log \rho(X_0^\beta) - \log p_{\mathrm{prior}}(X_T^\beta)$ and $\quad h_{u,\beta} := u \cdot \beta - \|u\|^2/2 - \nabla \cdot f$. The proof of likelihood formulation can be found in (Richter & Berner, 2024).

Based on the likelihood of path measures, diffusion models can be trained to sample from a given distribution by minimizing the following log-variance divergence (Richter & Berner, 2024)

$$D_{\mathrm{LV}}^{P_{\mathcal{X}^\beta}}(P_{\mathcal{X}^{\mathbf{u}}}, P_{\tilde{\mathcal{X}}}; \phi) := V_{P_{\mathcal{X}^\beta}} \left[ \log \frac{\mathrm{d}P_{\mathcal{X}^{\mathbf{u}}}}{\mathrm{d}P_{\tilde{\mathcal{X}}}} \right] \tag{10}$$

$$= V_{P_{\mathcal{X}^\beta}} \left[ D_{g_{u,\beta}} + S_u - R \right], \tag{11}$$

where $P_{\mathcal{X}^\beta}$ is a reference measure and $V[\cdot]$ is the variance function. The normalizing constant $Z$ is omitted safely since the variance function is shift-invariant.

## 4 METHODS

In this section, we present DiffEBM, our novel MCMC-free framework designed to jointly train a diffusion model and an energy-based model (EBM). The framework achieves this by minimizing the path divergence between the generative process and the reference process, while simultaneously maximizing the likelihood of the empirical data. We begin by describing the method for obtaining a well-calibrated diffusion model based on both empirical data and the energy-based model in Section 4.1. Subsequently, in Section 4.2, we outline how the learned diffusion model is employed to efficiently train the energy-based model. A comprehensive overview of the proposed framework is provided in Algorithm 1.

---

**Algorithm 1** DiffEBM Algorithm

---

**Input:** (1) Empirical data $\{x_i\}_i^n$; (2) Initial parameters $\phi$ and $\theta$ for diffusion model $u_\phi$ and energy function $E_\theta$ (3) hyper-parameters including weight coefficient $\eta$, $\alpha$ for diffusion model and EBM; number of training steps $K$; batch size $m$.

**Output:** Optimized parameters $\{\phi, \theta\}$

    **for** $k \leftarrow 1, \ldots, K$ **do**

        Sample empirical data $\{x_i^+\}_i^m \sim p_{\text{data}}(x)$

        Sample synthesized data $\{x_i^*\}_i^m$ by simulating the diffusion-based generator

        Given $\{x_i^+\}_i^m$, $\{x_i^*\}_i^m$ and $\theta$, compute $\mathcal{L}_{\text{G}}$ by following Equation 12

        Given $\{x_i^+\}_i^m$, $\{x_i^*\}_i^m$ and $\phi$, compute $\mathcal{L}_{\text{EBM}}$ by following Equation 14

        Optimize $\phi$ by following the gradient of $\nabla_\phi \mathcal{L}_{\text{G}}$

        Optimize $\theta$ by following the gradient of $\nabla_\theta \mathcal{L}_{\text{EBM}}$

    **end for**

---

## 4.1 LEARNING DIFFUSION-BASED GENERATORS

To effectively train a diffusion model capable of guiding the learning of an energy-based model (EBM) without relying on MCMC teaching, it is crucial that the diffusion model both accurately captures the underlying data distribution and aligns with the evolving EBM.

To meet the first objective, we employ denoising score matching, as formulated in Equation 7, ensuring that the diffusion model progressively learns to represent the empirical data distribution with high fidelity. For the second objective, we introduce the use of log-variance divergence as a training criterion to align the generator with the dynamic EBM distribution—a more complex challenge compared to training with a fixed, known target distribution.

The final learning objective for the diffusion model is formalized as follows:

$$
\begin{aligned}
\phi^* &= \arg\min_\phi \mathcal{L}_G(\phi) \\
&= \arg\min_\phi \left\{ (1-\eta)\mathcal{L}_{\text{DSM}} + \eta\, D_{\text{LV}}^{P_{\mathcal{X}^\beta}} \right\} \\
&= \arg\min_\phi \left\{ (1-\eta)\mathcal{L}_{\text{DSM}} + \eta\, V_{P_{\mathcal{X}^\beta}}\left[ D_{g_{\phi,\beta}} + S_\phi - R \right] \right\}
\end{aligned}
\tag{12}
$$

Here, $\eta \in [0,1]$ is a hyperparameter that balances the two learning objectives, and $R(X^\beta) := -E_\theta(X_0^\beta) - \log p_{\text{prior}}(X_T^\beta)$, where $X_T^\beta \sim p_{\text{prior}}$.

The learning process of the generator in our framework is MCMC teaching-free. When updating the generator to match the evolving EBM, we only need to generate samples from the current generator without requiring MCMC revisions. Additionally, we stop the gradient flow from the energy function when updating the generator's control function $\phi$, which is achieved by employing the detach operation in PyTorch.

## 4.2 LEARNING ENERGY FUNCTIONS

In each iteration, once the diffusion model is updated to align with the current EBM, the generator model is able to generate fare synthetic samples,which can be used to update EBM based on the MLE principle without relying on non-convergent short-run MCMC revisions. Specifically, we substitute the samples in Equation 3 with those generated via the generator as follows:

$$
\nabla_\theta \mathcal{L}_{\text{MLE}}(\theta) = \frac{\partial}{\partial \theta} \left( \frac{1}{n} \sum_{i=1}^n E_\theta(x_i^+) - \frac{1}{m} \sum_{i=1}^m E_\theta(x_i^*) \right),
$$

where $\{x_i^*\}_{i=1}^m$ are samples generated from the current learned diffusion model.

Since the diffusion model is also trained to maximize the likelihood of the empirical data via score matching, the learning of EBM can be significantly enhanced if EBM can be simultaneously trained to align with the current diffusion models. Moreover, MCMC teaching can be unstable in practice, as two distributions of the generator and the EBM tend to chase each other,potentially leading to

un-convergence. To mitigate the risk of excessively large gradient updates during training the EBM under the MLE criterion, the following objective function is proposed to ensure that the updated EBM would not diverge drastically from the current diffusion model, allowing the EBM to effectively learn from the current generator if the diffusion model we trained is to maximize the empirical data likelihood.

$$
D_{\mathrm{LV}}^{P_{\mathcal{X}^\eta}}\left(P_{\mathcal{X}^\phi}, P_{\bar{\mathcal{X}}}; \theta\right) \coloneqq V_{P_{\mathcal{X}^\eta}}\left[\log \frac{\mathrm{d}P_{\mathcal{X}^\phi}}{\mathrm{d}P_{\bar{\mathcal{X}}}}\right] \tag{13}
$$

$$
= V_{\mathcal{X}^\eta}\left[R\right] - 2\mathbb{E}_{\mathcal{X}^\eta}\left[AR\right] + 2\mathbb{E}_{\mathcal{X}^\eta}\left[A\right]\mathbb{E}_{\mathcal{X}^\eta}\left[R\right] + \text{const.},
$$

where $\mathcal{X}^\eta$ is a reference process, $A \coloneqq D_{g_{\phi,\eta}} + S_\phi$ and $\quad R(X^\eta) \coloneqq -E_\theta(X_0^\eta) - \log p_{\mathrm{prior}}(X_T^\eta)$. The proof relies on the previously introduced definition of likelihood and a straightforward application of the variance property, given by $\mathrm{Var}[A] = E[A^2] - (E[A])^2$."

The final learning objective for the energy-based model to match the current generator and the empirical distribution can be described as follows:

$$
\boldsymbol{\theta}^\star = \arg\min_{\boldsymbol{\theta}} \mathcal{L}_{\mathrm{EBM}}(\theta)
$$

$$
= \arg\min_{\boldsymbol{\theta}} \{(1 - \alpha)\mathcal{L}_{\mathrm{MLE}} + \alpha\, D_{\mathrm{LV}}^{P_{\mathcal{X}^\eta}}\}, \tag{14}
$$

where $\alpha \in [0, 1]$ is a hyperparameter to balance two objective functions. Similarly, we stop the gradient of the control function $\phi$ of the generator when learning the energy function, which can be done by using the detach operation in PyTorch.

## 5 EXPERIMENTS

In this section, we validate the effectiveness of our proposed method on several benchmarks in density estimation and generative modeling. Moreover, we show that our method can learn valid energy functions that result in meaningful samples even using a long-run MCMC.

### 5.1 EXPERIMENTAL SETUP

**Baselines** We compare our proposed method with several classical EBM learning algorithms, including MCMC sampling-based MLE method(MCMC) (Liu & Liu, 2001; Younes, 1999), denoising score matching (DSM) (Vincent, 2011) and deep energy estimator networks (DEEN) (Saremi et al., 2018).

**Setup** The architecture of the energy function is composed of three $1 \times 1$ convolutional layers with the SiLU activation function (Elfwing et al., 2018). The encoder of the score function is a Fourier multilayer perceptron (MLP) model (Zhang & Chen, 2022). We use the Prodigy optimizer with the hyperparameters listing in Appendix. The same set of hyperparameters is used in our method for all tested benchmark problems. We use the variance-preserving SDE for the reference process with a linear scheduler as $\beta(t) = \beta_{\min} + t(\beta_{\max} - \beta_{\min})$ where $\beta_{\max} = 10$ and $\beta_{\min} = 0.1$ for all tested problems.

**Dataset** We use five classical simulation-based datasets to demonstrate the performance of our algorithms, including two Gaussian mixture datasets with ground truth likelihood functions, Swissroll, Checkerboard and 2spirals.

**Evaluation** The Sinkhorn distance (Cuturi, 2013) is used to evaluate the performance of different algorithms. It is an entropy regularized version of the Wasserstein distance between the synthetic distribution and the real data distribution. The average Sinkhorn distance is reported in Table 1 by using 10 different seeds.

### 5.2 DENSITY ESTIMATION

We begin by conducting experiments to demonstrate that our proposed method can learn the meaningful energy function on two Gaussian mixture model (GMM) datasets with known density functions. The probability density of the eight GMM dataset (8Gaussians) is defined as

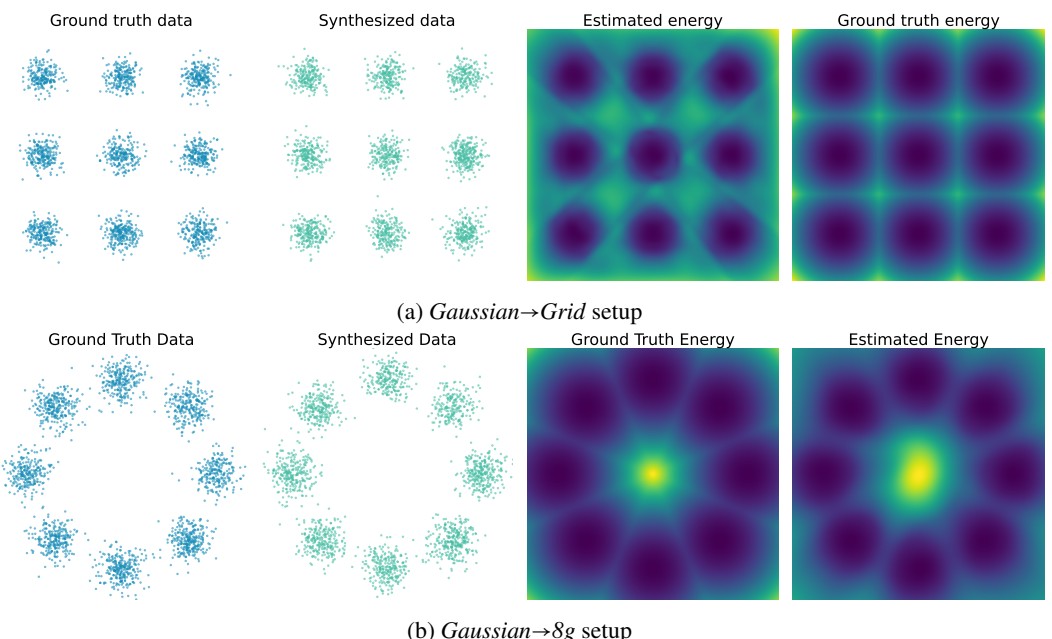

(a) *Gaussian→Grid* setup

(b) *Gaussian→8g* setup

Figure 1: Validation of our method in density estimation and generative modeling on *Gaussian→8Gaussians* and *Gaussian→Grid* setup. The ground truth energy function in analytically known for the GMM dataset. The synthesized data is obtained by simulating the diffusion model starting from Gaussian noise.

$p(x) = \frac{1}{8} \sum_{i=1}^{8} \mathcal{N}\left(x; \mu_i, \mathbb{I}\right)$, where $\mu_i = \left(\sqrt{2} \cdot \cos\left(\theta_i\right), \sqrt{2} \cdot \sin\left(\theta_i\right)\right)$ and $\mathbb{I}$ denotes identity matrix. The dataset Grid consists of a GMM with nine modes, arranged in a $3 \times 3$ grid. Each mode is centered at coordinates derived from an evenly spaced grid over the range $[-5, 5]$ along both the $x$-axis and $y$-axis. The probability density of Grid is defined as $p(x) = \frac{1}{9} \sum_{i=1}^{9} \mathcal{N}\left(x; \mu_i, \mathbb{I}\right)$, where $\mu_i \in \left\{(x_i, y_j) \mid x_i, y_j \in \{-5, 0, 5\}\right\}$ are positioned on a $3 \times 3$ grid. The covariance matrix $\Sigma = \sigma^2 \mathbb{I}$, where $\sigma = \sqrt{0.3}$, defines the spread of each Gaussian component.

We visualize the training data, synthetic samples, the ground truth density and the learned energy function for two GMM datasets in Figure 1. Figure 1 shows that our method does not suffer from the mode collapse issue and can capture all the modes, demonstrating that our proposed method performs well both as a valid sampler and an effective energy function estimator.

## 5.3 GENERATIVE MODELING

We validate the generative modeling ability of our proposed method on three datasets including 2spirals, Swissroll and Checkerboard. We visualize the real dataset, the generated samples from the trained diffusion models and the estimated energy function in Figure 2. We can see from Figure 2 that our joint training the generator and the EBM can learn an effective generative model that can generate more samples similar to the given empirical dataset.

## 5.4 COMPARISONS TO THE BASELINES

We demonstrate the comparison of the learned energy function with several baselines in Table 1, which summarizes the Sinkhorn distance that measures the discrepancy between the synthesized data samples and the ground truth data. Table 1 shows that our method is effective in generating realistic data. Figure 3 shows that our method can learn better energy function when comparing with the score matching.

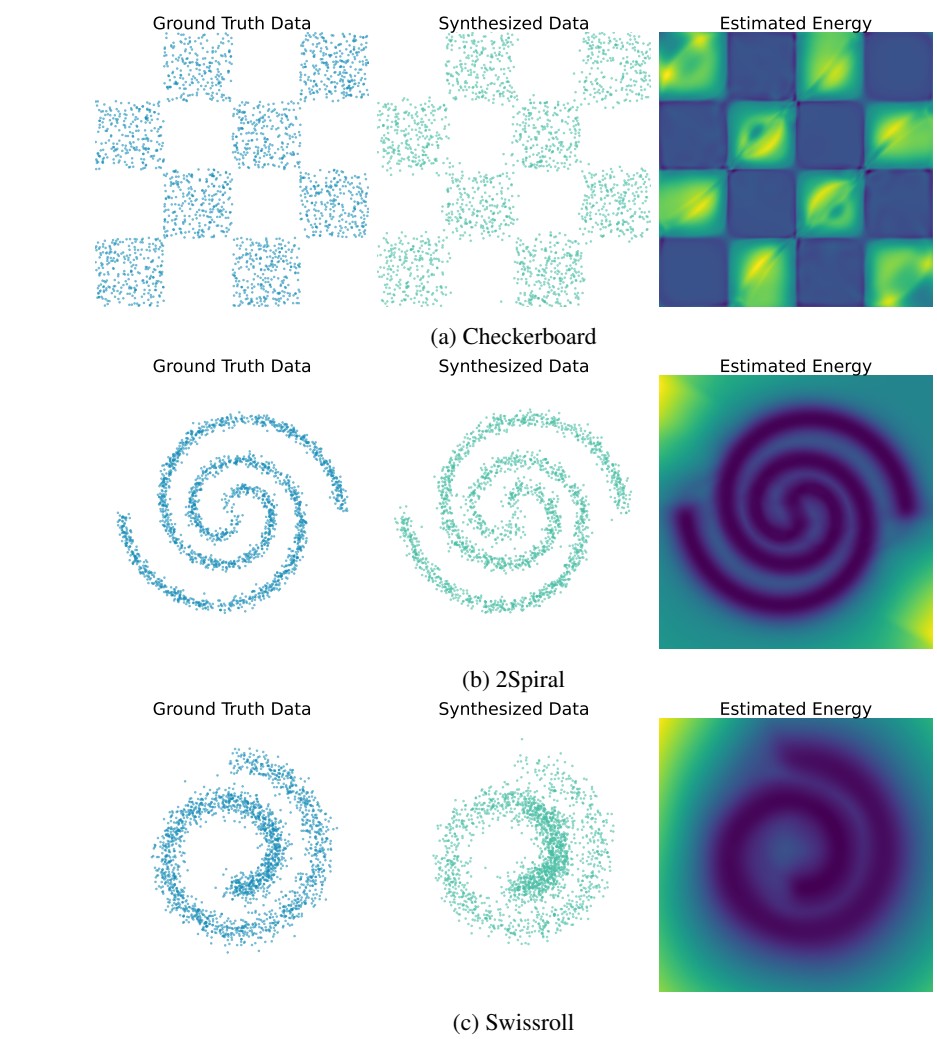

Figure 2: Validation of our method in generative modeling on Swissroll, 2Spirals and Checkerboard. The synthesized data is obtained by simulating the diffusion model starting from Gaussian noise.

Table 1: Performance comparison of different algorithms on benchmark problems in terms of the Sinkhorn distance.

| Methods | Swissroll | 2Spirals | Checkerboard |
|---|---|---|---|
| MCMC (Liu & Liu, 2001; Younes, 1999) | 0.31066 | **0.12194** | 0.43577 |
| DEEN (Saremi et al., 2018) | 0.12555 | 0.33339 | 0.74726 |
| DSM (Vincent, 2011) | 0.13140 | 0.32821 | 0.73577 |
| Ours | **0.12147** | 0.28954 | **0.37430** |

## 5.5 LIMITATIONS AND FUTURE WORK

One major limitation of the diffusion-based sampling is we need to simulate from the diffusion models, which is time-consuming for high-dimensional data. Our future work is to develop fast sampling method for learning EBM for modeling high-dimensional data.

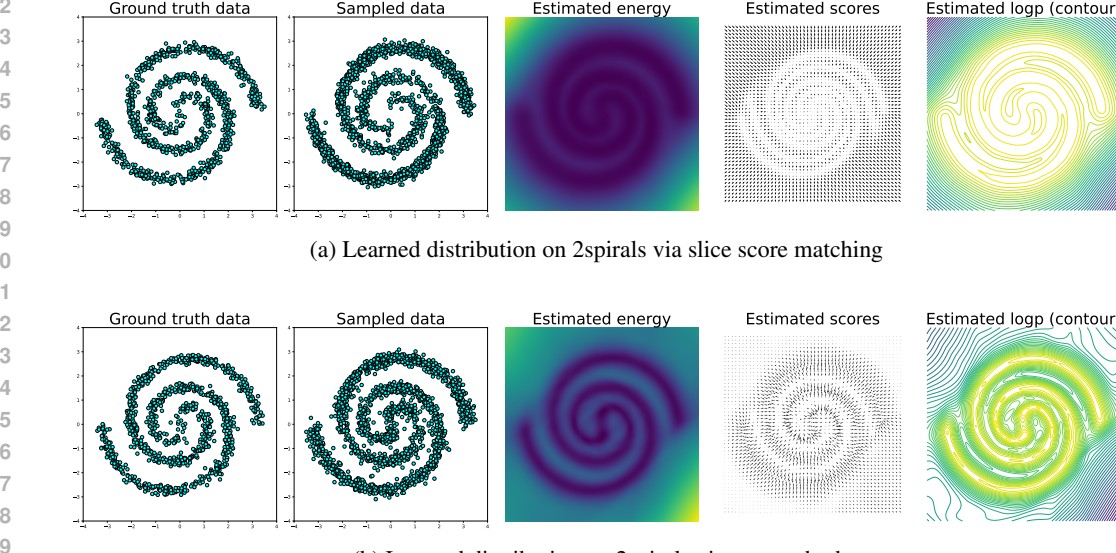

(a) Learned distribution on 2spirals via slice score matching

(b) Learned distribution on 2spirals via our method

Figure 3: Results on 2spirals via slice score matching. As we can see from the last two plots, the learned energy function is not very well, although the generated samples seem great (the second plot). Our method has better learned energy function. However, there are many noisy fake samples for our method (working on it).

## 6 CONCLUSION

In this paper, we propose a novel MCMC teaching-free framework to learn energy-based models. We leverage the variational formulation of divergence between time-reversed diffusion paths to jointly train an EBM and a diffusion-based generative model. By training the generator and the EBM to match with each other, the samples simulated from the generator can be viewed directly as fair synthesized data to update EBM via MLE, and the refined EBM model can directly be used to update the generator based on the unnormalized density-guided path divergence, avoiding mode collapse and slowing mixing issues of traditional MCMC sampling-based EBM learning methods. We verify the effectiveness of the proposed method on the energy estimation, density estimation, score estimation, and the quality of synthetic data on simulations.

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
