# OpenReview forum: "No MCMC Teaching For me: Learning Energy-Based Models via Diffusion Synergy"
_ICLR.cc/2025/Conference — Submitted to ICLR 2025_

### Official Review · Reviewer_xpjM · 2024-10-31

**Soundness:** 2
**Presentation:** 2
**Contribution:** 2
**Rating:** 3
**Confidence:** 5

**Summary:**

The paper introduces a method for training Energy-Based Models (EBMs) without relying on Markov Chain Monte Carlo (MCMC). In each training step, a diffusion-based sampler is learned to match the current EBM and data distribution. This sampler is then used to generate samples, enabling maximum likelihood training of the EBM. Experimental results on synthetic toy data demonstrate the method's effectiveness.

**Strengths:**

The proposed method eliminates the need for MCMC. While it involves training an additional diffusion-based sampler, it avoids the bias issues associated with MCMC, provided the sampler is well-trained.

**Weaknesses:**

- The proposed method is evaluated solely on 2D synthetic data. Testing it on high-dimensional datasets, such as images, would help assess its scalability.
- There are some missing baselines:
    - Variational Inference: [1] propose to estimate the partition function using variational inference, which is also MCMC-free
    - Noise Contrastive Estimation (NCE) [2]. NCE is MCMC-free and can work very well on 2d density estimation.
    - Energy Discrepancy (ED) [3] is a recently introduced method for training EBMs without MCMC. It offers compelling theoretical guarantees and has demonstrated effectiveness in tasks like density estimation and image modelling.

[1] Duvenaud D, Kelly J, Swersky K, Hashemi M, Norouzi M, Grathwohl W. No MCMC for me: Amortized samplers for fast and stable training of energy-based models. InInternational Conference on Learning Representations (ICLR) 2021.

[2] Gutmann, Michael, and Aapo Hyvärinen. "Noise-contrastive estimation: A new estimation principle for unnormalized statistical models." *Proceedings of the thirteenth international conference on artificial intelligence and statistics*. JMLR Workshop and Conference Proceedings, 2010.

[3] Schröder, Tobias, et al. "Energy discrepancies: a score-independent loss for energy-based models." *Advances in Neural Information Processing Systems* 36 (2024).

**Questions:**

- The sampler is trained using a loss function designed to align the data distribution with the current EBM. While this approach is unbiased when the EBM is well-trained, it can lead to a biased maximum likelihood estimator if the EBM is underfitting, which is common in the early stages of training. It would be great to see how it works without the DSM loss in sampler training.
- The EBM is trained to match the data distribution and the current sampler. It would also be valuable to see the results when the sampler matching loss is omitted during EBM training.

---

> ### Author Response · Authors · 2024-11-26
> **Thanks for your review**
>
> Dear Reviewer,
>
> Thank you for your insightful review. We greatly appreciate your feedback and will include additional experiments on high-dimensional datasets, as well as comprehensive comparisons with the suggested baselines. Below, we address your specific points in detail:
>
> - On training the sampler without the DSM loss
>
>   Based on our experience, it is possible to train the diffusion sampler entirely without the DSM loss, using only the log-variance to update the sampler while keeping the EBM distribution fixed. However, on high-dimensional datasets, the log-variance loss can be computationally inefficient due to the need to simulate from the diffusion models. We aim to explore and develop more efficient methods to address this limitation in the future.
>
> - On omitting the sampler matching loss during EBM training
>
>   Since we mainly use the MLE principle to train the EBM, it is necessary for the fake samples to align with the EBM distribution, making the sampler matching loss essential. One exception occurs when the diffusion models are not updated via the DSM loss but are instead solely updated to match the sampler. We will provide more experimental results to further explore this scenario in the revised version.
>
> We hope these clarifications address your concerns and look forward to incorporating your suggestions to improve our work.

---

### Official Review · Reviewer_7Qot · 2024-11-03

**Soundness:** 2
**Presentation:** 2
**Contribution:** 2
**Rating:** 3
**Confidence:** 4

**Summary:**

The authors propose to replace the traditional MCMC sampling for learning energy-based models (EBMs) with sampling from diffusion models. Generation speed and sample quality are major bottlenecks in learning EBMs, and the experiments show part of those problems are addressed. The used sampling method from EBMs is not novel, as it follows the method from recent work by Richter & Berner (2024). While the proposed method is straightforward and reasonable, its contribution is incremental.

**Strengths:**

The proposed method is reasonable.

**Weaknesses:**

The contribution is not significant. It merely incrementally extends the published sampling method to learning EBMs. If the authors could address a major challenge in applications using the diffusion sampling, the contribution would be more noteworthy.

Minor comment:
Although the equations (8) through (11) were borrowed from previous literature, the authors have to explain those equations in their own words. The provided explanation regarding the diffusion sampling from previous work does not clarify why the proposed sampling should be better than MCMC.

**Questions:**

The paper is clearly written.

---

> ### Author Response · Authors · 2024-11-26
> **Thanks for your  review**
>
> Dear Reviewer,
>
> Thank you for your insightful review. We appreciate your feedback and will clarify the effectiveness of the proposed method in the revised version. We would also like to address the following points for further clarification:
>
> - Our method is not an incremental approach that merely "extends the published sampling method to learning EBMs." If one were to simply extend existing sampling methods, the MCMC-teaching strategy could be adapted for learning EBMs. In contrast, we propose a novel MCMC-teaching-free approach to learning EBMs by minimizing the log-divergence between path measures, which, to the best of our knowledge, has not been explored for this purpose.
>
> - Learning EBMs is inherently an application of diffusion-based sampling. However, we will include more downstream applications of the learned EBMs in the revised version.
>
> - It is well-known that MCMC struggles in practice when applied to learning EBMs. We will provide a more detailed explanation of why diffusion-based sampling offers advantages over MCMC in this context.
>
> We hope these clarifications address your concerns, and we will ensure that the revised version fully reflects these points.

---

### Official Review · Reviewer_p1Jc · 2024-11-04

**Soundness:** 3
**Presentation:** 3
**Contribution:** 2
**Rating:** 3
**Confidence:** 4

**Summary:**

This paper presents a novel approach to training Energy-Based Models (EBMs) without relying on Markov Chain Monte Carlo (MCMC) methods, which are traditionally used but can be unstable and biased in high-dimensional settings. The proposed method, referred to as DiffEBM, employs a diffusion-based framework that trains an EBM and a diffusion model simultaneously, effectively eliminating the need for MCMC by leveraging divergence in time-reversed diffusion paths.

The paper identifies core limitations of MCMC, such as mode collapse and slow mixing, which hinder EBM training. To address these, DiffEBM introduces an objective function to match the EBM and diffusion model, using samples from the latter as unbiased approximations of the data distribution, sidestepping the biases associated with short-run MCMC. The diffusion model is trained using the technique proposed in [Richter & Berner, 2024]. In contrast, the EBM is updated based on synthesized data generated by the diffusion model.

Experimentally, DiffEBM demonstrates superior performance on various benchmarks, including Gaussian mixture datasets and synthetic data distributions like 2Spirals and Swissroll. Performance is evaluated using Sinkhorn distance to compare generated samples to ground-truth distributions.

In summary, DiffEBM introduces a diffusion-driven training framework for EBMs that enhances efficiency, stability, and sample fidelity by removing MCMC-based sampling, thus providing an alternative pathway for EBM training in complex generative tasks​

**Strengths:**

The paper's primary strength lies in its innovative approach to training Energy-Based Models (EBMs) without the reliance on Markov Chain Monte Carlo (MCMC) methods, which have known limitations in high-dimensional contexts. Traditional MCMC-based EBM training often suffers from mode collapse, slow mixing, and biased samples, especially with short-run MCMC. By introducing a diffusion-based generative model that jointly trains with the EBM, the authors successfully bypass these challenges. This joint training, which uses divergence between time-reversed diffusion paths as an objective function, eliminates the need for MCMC teaching. As a result, DiffEBM achieves higher sample quality by aligning the generative model directly with the EBM’s learned distribution, making it a valid alternative to MCMC-based methods.

**Weaknesses:**

The method proposed in this paper, while innovative, introduces significant computational demands that undermine its practical efficiency. The core idea—training an EBM in tandem with a diffusion-based generative model to avoid the pitfalls of MCMC sampling—replaces the complexity of MCMC with an equally demanding requirement: learning a second, paired generative model that must be iteratively updated alongside the EBM. This approach involves repeatedly sampling from the diffusion model during each training step, as highlighted in Algorithm 1, line 223, where a full sequence of diffusion sampling is performed at each iteration. This reliance on diffusion sampling makes the process computationally intensive, as each update to the EBM requires a costly simulation of the diffusion process to produce high-fidelity samples, compounding the training time considerably. Moreover, the iterative nature of sampling across the full diffusion chain can easily lead to instability, especially if the parameters of the generative model diverge from the EBM, creating an oscillating learning dynamic that may fail to converge.

Another key issue arises from the purpose of training the EBM when the diffusion model, a high-capacity generative framework in its own right, is already optimized to produce accurate samples. If the diffusion model alone can capture the empirical data distribution effectively, as evidenced in the quality of generated samples, the rationale for learning an additional EBM becomes questionable. The diffusion model could theoretically fulfill the generative modeling objective by itself, rendering the EBM redundant for many practical applications. Training both models in parallel may not yield substantial benefits over simply using the diffusion model, especially given the EBM’s limited advantage in scenarios where the diffusion model is already well-aligned with the data distribution. Thus, while the framework’s goal is to leverage the EBM’s interpretability and robustness in capturing complex energy landscapes, the computational cost and redundancy associated with dual-model training suggest a misalignment between the theoretical motivation and the efficiency of the method.

Another limitation is the lack of direct comparison with standalone diffusion-based generative models, which would offer a fairer baseline for evaluating the proposed approach. Since the method relies heavily on a diffusion model, comparing it against established diffusion-only schemes—or even against samples generated solely by its own diffusion model—would help clarify whether the added complexity of training an EBM provides real benefits. Without such comparisons, it’s uncertain if the dual-model approach improves performance significantly over simpler, diffusion-based methods alone, potentially overestimating its effectiveness.

Finally, in my opinion the considered datasets are too simplistic to claim that the proposed method really has superior performance compared to other schemes.

**Questions:**

Given the substantial computational load and potential instability introduced by training an EBM alongside a diffusion model, have you considered alternative strategies to reduce the computational demands, such as truncated or approximate diffusion sampling, without compromising sample quality?

Since your approach integrates a high-capacity diffusion model, could you clarify the unique advantages of training an EBM in tandem? Specifically, how does the EBM contribute to the overall performance compared to using the diffusion model alone for generative tasks?

To better understand the value of the dual-model approach, would you consider evaluating your method on more complex datasets and comparing it directly against standalone diffusion-based generative models, as well as using samples from your diffusion model as a self-baseline? This would help clarify any performance gains provided by the EBM, particularly on challenging, high-dimensional data where diffusion-only methods may already perform well.

Can the authors clarify what they mean in the experimental section when they refer to Denoising Score Matching? What is the relationship with the sliced score matching mentioned in Figure 3?

Also, a minor point, why in Fig 3 are the ground truth samples different for the two methodologies?

---

> ### Author Response · Authors · 2024-11-26
> **Thanks for your review**
>
> Dear Reviewer,
>
> Thank you for your insightful feedback. We greatly appreciate your suggestions and will incorporate them to enhance our work. Specifically, we plan to develop a more efficient version of the proposed method, conduct additional experiments on high-dimensional datasets, and perform comprehensive comparisons with diffusion models to further validate the effectiveness of our approach.

---

### Meta-Review · Area_Chair_dSaT · 2024-12-18

**Metareview:**

The paper proposes a method for training energy based models without using MCMC. While the problem the paper addresses is very relevant and important, all three reviewers agreed that the paper should be rejected from ICLR. The computational demands that it introduces works to undermine its practical efficiency, which is a criticism of MCMC to being with. Also noted was that higher dimensional examples such as images would help make the method more compelling.

**Additional Comments On Reviewer Discussion:**

There wasn't much discussion between the authors or reviewers and no opinions were changed by the brief rebuttal.

---

### Decision · Program_Chairs · 2025-01-22

Reject